# Nature and type of patient-reported safety incidents at a tertiary hospital in South Africa during the COVID-19 period (2018–2021)- A retrospective review

Swabhavika Singh[1☯], Ozayr Mahomed [1,2☯]*

**1** Discipline of Public Health Medicine- University of KwaZulu Natal, Durban, South Africa, **2** Dasman Diabetes Institute, Kuwait City, Kuwait

☯ These authors contributed equally to this work.
* mahomedo@ukzn.ac.za

**Data Availability Statement:** All relevant data are within the paper and its Supporting information files.

## Abstract

The healthcare sector in South Africa has seen a rise in medical errors and negligence adversely affecting healthcare delivery. This study aimed to determine the nature and magnitude of Patient Safety Incidents at a tertiary hospital between April 2018 to March 2021 (the COVID-19 era). A retrospective review of Patient Safety Incidents forms and clinical records of inpatients above 12 years with a reported incident were included. The overall incidence of patient safety incidents was 4.40 PSIs per 10 000 patient-days. The majority of PSIs were harmful incidents (An incident that results in harm to a patient that is related to medical management, in contrast to disease complications or underlying disease) at 72.50% [145/200], followed by no harm incidents at 18.00% [36/200] and 9.50% [19/200] near-miss incidents. The five highest incident types were clinical process/procedure [83/200; 41.50%], behaviour [49/200; 24.50%], falls [28/200; 14%], complications [20/200; 10%], and pressure sores [11/200; 5.50%]. Incidents occurred more frequently within the week (83.50%) and during day shift (67.50%). Overall, the study showed low PSI reporting rates which is an inherent challenge with voluntary reporting. Notably, there has been an increasing trend in reporting. The results reflect a reporting culture more inclined to bring awareness to incidents that have negative outcomes.

## Introduction

Globally, patient harm occurs in 10% of all people using a healthcare facility. Patient harm is ranked fourteenth in the list of the global burden of diseases. Notably, 80% of these Patient Safety Incidents (PSIs) are preventable [1]. The World Health Organization (WHO) estimates that 42.7 million PSIs occur every year as a result of 421 million hospitalisations. In low and middle-income countries (LMICs) it is estimated that 134 million PSIs occur, resulting in 2.6 million deaths [2]. These incidents directly impact patients, the healthcare system, and the economy [3]. In 2016 in the United Kingdom (UK) the National Institute of Health Research announced that a PSI is reported once every 35 seconds, with approximately 3.6% of deaths in

**Funding:** The author(s) received no specific funding for this work.

**Competing interests:** The authors have declared that no competing interests exist.

**Abbreviations:** ICU, Intensive care unit; KZN, KwaZulu Natal; LMIC, Low- middle-income country; PSI, Patient Safety Incident; SAC, Severity Assessment Code; UK, United Kingdom; US, United States of America; WHO, World Health Organization.

acute hospital settings secondary to PSIs being avoidable [4]. In 2011 the WHO published a study of six Middle Eastern and two African countries, one of which was South Africa. The study was a retrospective medical record review of over 18 000 patients from 26 hospitals identifying patient safety incidents. The study showed that almost a third of patients in which harm was identified died, while 14% had a permanent disability, 16% a moderate disability, and the remainder were classified as minimal (30%) or non-specified (8%). Most incidents (80%) were preventable [5]. Common types of patient safety incidents that occur globally include nosocomial infections; incorrect use or administration of medication; adverse drug reactions; procedure-related errors with a greater burden in surgically related fields; misdiagnosis or poor clinical management or care practices resulting in pressure sores; harm from falls; and faulty equipment [6, 7].

The WHO stated that the burden of PSIs is disproportionately greater in LMICs, accounting for two-thirds of harm events [7]. Attributing factors include poor healthcare systems from resource limitations, poor infrastructure, insufficient health workforce and access to care [8]. Conversely, the effects of PSIs negatively impact the healthcare system. Within the healthcare system, approximately 15% of total hospital costs are attributable to PSIs [9].

The prevalence of patient harm and its effects on patients, the healthcare system, and the economy are profound. The negative impact suffered from these adverse outcomes results in loss of life or disability and financial fallouts, creating a culture of patient distrust and a litigious society. In addition, patients bear the brunt in loss of lives, permanent disability, loss of income, psychological distress, fear of retribution, and growing mistrust in the institution during the most vulnerable times of their lives [10]. In South Africa, the resource-constrained healthcare sector is further burdened by a rise in litigation secondary to medical negligence, as the state bears vicarious liability [11].

The WHO has identified patient safety as a key strategic priority for universal healthcare calling on member states to take actions directed at decreasing medical harm and negligence by creating an enabling environment that respects the dignity of patients, provides safe and effective patient care, and reduces litigation costs [12]. The WHO has called on member states to undertake research that assesses the nature and magnitude of patient safety incidents and reporting behaviour [6, 13].

In the local context there is a gap relating to the magnitude, nature and factors contributing to patient safety incidents. The current facility is a bustling tertiary-level hospital with a high patient turnover, providing a myriad of services across the clinical spectrum. As an academic institution, it supports improvement through learning. Undertaking an analysis of this data will also add to local epidemiological information; and contextualise the burden PSIs. Thus quality improvement and learning areas for hospital management, stakeholders, and policymakers will be identified.

The aim of this study was to determine the incidence, and nature of patients safety incidents reported between 2018 and 2021.

## Methods

### Study design, and setting

A retrospective cross-sectional observational study was conducted at a tertiary-level academic hospital in the eThekwini district of KwaZulu-Natal affiliated with the University of KwaZulu-Natal's Nelson R. Mandela School of Medicine. The hospital has a capacity of 859 beds with a turnover of approximately 22 000 patients per month. While categorised as a tertiary level service, the hospital receives patients directly from some primary and community healthcare clinics with trauma, surgical, and obstetric problems requiring higher level of care.

## Study population

The study population included all patients above the age of 12 years who were admitted to between 1st April 2018 and 31st of March 2021 and had a patient safety incident reported were included as the study population. No sampling was conducted as the study intended to have a full census of all PSIs within this period to provide a complete analysis. All outpatients and staff-related incidents were excluded.

## Data collection tool and procedure

The main source document for the data were the patient safety incident notification forms and patient safety incident registers. These are standardised forms as per the South African National Guidelines for Patient Safety Reporting and Learning that were published in 2017. The current reporting format at the hospital is in the paper format. The South African Guidelines adopt the Minimal information Model as proposed by the World Health Organisation[14]. Any unplanned or unintended event or circumstance that could have resulted or did result in harm to a patient while in the care of a health facility is required to be reported [14]. According to the Guidelines Serious Adverse Events should be prioritised. The tool is structured to collect the following information:

- Incident identification o patient (a person who is a direct or indirect recipient of healthcare and involved directly or indirectly in the PSI); time (date and time of day when the incident occurred); location (physical environment in which a PSI occurs)

- Contributing factor (factor with the potential to cause harm. It refers to the product, device, person, or any elements involved in the incident with the potential to influence it)

- Incident type (a descriptive term for a category made up of incidents of a common nature, grouped because of shared, agreed features)

- Incident outcomes (all impacts upon a patient or an organisation wholly or partially attributable to an incident)

- Resulting actions (identify immediate or indirect action taken that relates to the patient or the organisation to improve the situation or prevent the reoccurrence of an incident)

- Reporter (person who collects and writes information about the incident)

Root cause analysis should be conducted by the patient safety committee within the hospital. At a unit level, the team comprises the clinical head of department, nursing unit manager, quality assurance manager, infection prevention control practitioner and pharmacist.

Information was obtained from the monitoring and evaluation department at the Hospital who are responsible for collecting, collating, and reporting the patient safety incidents, and conducting and recording meetings. The total number of inpatients for the research period was obtained. The health information system department provided data on total hospital admissions during the period.

A secondary data extraction sheet was used to obtain the following information pertaining to the patient safety incident: Patient factors: Patient identification, age, gender, diagnosis, Chronology of events: Month, day, and time the PSI occurred; date reported; number of days to report Severity Assessment code location, and discipline; Description of PSI: Type (no harm; near miss; harmful) and classification according to incident type (10 sub-categories);; Outcomes: Patient (none; mild; moderate; severe; death) and organizational (damage to property, increased resources, media attention, complaint, disrepute, legal ramifications, and other).

## Data analysis

All the data was entered initially into Microsoft Excel 2016 and then imported into IBM statistical software for data science (STATA). All data was on a password-protected personal computer with an installed antivirus. Since the aim of this study was to provide epidemiological data a descriptive univariate analysis of data was conducted to summarise patient safety incidents. The patient characteristics (age and gender); incident characteristics (location, discipline, month, day and time); PSI type (no harm, near miss, harm); contributory factors (staff, patient, environment, and organisational); and PSI outcomes for both patient and facility will be described. Categorical data were expressed as frequencies and percentages. Frequency distribution tables will be used to illustrate the data. The proportion of PSIs were calculated by the total number of PSIs versus the total hospital admissions over the period and the total inpatient days.

## Results

There was a total of 46 707 adult admissions during this time period, totalling 454 684 inpatient days. The average length of stay was 6.3 days; and the average bed occupancy was 64% (Table 1). There was a total of 265 patient safety incidents from 1$^{st}$ April 2018 to 31$^{st}$ March 2021. Of these, 243 were inpatient related, of which 4 were duplicated. There were 39 incidents that involved those aged twelve years or younger. A final sample size of 200 patient safety incidents was included in this study.

### Incidence of patient safety incidents

There was a total of 4.40 PSIs reported per 10 000 patient-days over the three-year period. The number of reported incidents per year has progressively increased over the study period from 22 (1.37 PSIs per 10 000 patient-days) in Year One To 68 (4.21 PSIs reported per 10 000 patient-days) in Year Two and to 110 (8.33 PSIs reported per 10 000 patient-days) in Year Three (Table 1). The overall percentage of admissions with at least one incident reported was 0.43% over the three years. There was an increasing trend from Year One at 0.15%, Year Two at 0.46% and Year Three at 0.64%. Reporting increased by three times from Year One to Year Two and increased by 1.4 times from Year Two to Year Three.

### Type of patient safety incidents

Patient safety incidents are divided into three categories: harm, no harm, and near miss. The majority of PSIs were harmful incidents (An incident that results in harm to a patient that is related to medical management, in contrast to disease complications or underlying disease) at

**Table 1. The annual inpatient data from 1$^{st}$ April 2018 to 31$^{st}$ March 2021.**

|  | 1 April 2018–31 March 2019 | 1 April 2019–31 March 2020 | 1 April 2020–31 March 2021 |
|---|---|---|---|
| *Bed Occupancy* | 62% | 73% | 58% |
| *Average length of stay* | 6,6 | 6,4 | 5,9 |
| *Total inpatient Days* | 161 110 | 161 462 | 132 112 |
| *Total admissions* | 14 660 | 14 841 | 17 206 |
| *Number of PSI* | 22 | 68 | 110 |
| *PSI per 10000 patient -days* | 1,37 | 4,21 | 8,33 |
| *Percentage of admissions with PSI* | 0,15% | 0,46% | 0,64% |

72.50% [145/200], followed by no harm incidents at 18.00% [36/200] and 9.50% [19/200] near-miss incidents. The majority of PSIs (64% [128/200]), were scored as severity assessment code (SAC) two, and followed by SAC 3 at 28% [56/200]. Only 8% [16/200] were classified as SAC 1-serious incidents.

## Incident characteristics

**Patient characteristics.** The gender distribution of PSIs showed a slightly higher number of females at 53% [106/200] versus males at 47% [94/200] (Table 2). Based on the patient safety incident type, harm-related incidents were highest amongst females at 53.79% [78/145] versus males at 46.21% [67/145]. No-harm or near-miss-related PSI incident types accounted for 49.09% [27/ 55] of females, and 69.09% [38/ 55] of males. There was no significant association between gender and type of PSI (Table 1).

The mean age [N = 200] was 37.84 years (SD 14.80 years), with a median age of 34 years (IQR 27.5–45.5 years). Most PSIs reported were amongst those between the ages of 30–44 years [42%; 88/200] (Table 2). The total number of obstetric-related PSIs were 44; and removal of these resulted in a mean age of 40.10 (SD 15.73) and median age of 36 (IQR 28–50). There was a slightly higher number of incidents reported in those who were below the median age at 53% [106/200]. Harm incidents reported at or below the median age were 49.66% [72/145]; and those above the median age were 50.34% [73/145]. No-harm and near-miss incidents reported at or below the median age were 61.82% [34/55]; and those above the median age were at 38.18% [21/55]. There was no significant association between patients safety incidents and patient age.

**Location and discipline.** The highest number of PSIs were allocated to a general ward at 62.50% [125/200] followed by the operating theatre at 32.50% [65/200]. The remaining incidents occurred in the adult intensive care unit at 3% [6/200] and 2% [4/200] in acute medical admissions. The highest reported PSIs were from the obstetrics and gynaecology discipline 28% [56/200]. The orthopaedic and medical departments had 21.50% [43/200] each. The surgical discipline reported 11% [22/200] and anaesthetics and psychiatry reported 7.50% [15/200] each. The intensive care discipline reported only 3% [6/200] over the study period.

**Time of occurrence of PSI.** Patient safety incidents reporting was slightly higher in the first six months at 51.50% [103/200] compared with the latter half of the year, at 48.50% [97/ 200]. The no-harm and near-miss related incidents that occurred during the first six months

**Table 2. Univariate analysis of patient factors and time related variables versus type of PSI between 1st April 2018 and 31st March 2021.**

| Variable (N = 200) | Sub-category | Near miss or no harm | Harm | Total population | Chi squared | Unadjusted odds ratio | CI Interval | p value |
|---|---|---|---|---|---|---|---|---|
| *Gender* | Female | 27 (14%) | 78 (39%) | 105 (53%) | | Comparison group | | |
| | Male | 28 (14%) | 67 (34%) | 95 (48%) | 0,35 | 0,83 | 0,42–1,61 | 0,55 |
| *Age* | 15–29 | 20 (10%) | 45 (23%) | 65 (32%) | 4,83 | Comparison group | | |
| | 30–44 | 27 (14%) | 57 (29%) | 84 (42%) | | 0,94 | 0,47–1,89 | 0,86 |
| | 45–59 | 5 (3%) | 27 (14%) | 32 (16%) | | 2,4 | 0,86–7,14 | 0,12 |
| | ≥ 60 years | 3 (2%) | 16 (8%0 | 19 (10%) | | 2,37 | 0,62–9,06 | 0,21 |
| *Month of incident* | January- June | 29 (15%) | 74 (37%) | 103 (52%) | | Comparison group | | |
| | July- December | 26 (13%) | 71 (36%) | 97 (49%) | 0,046 | 1,07 | 0,55–2,09 | 0,83 |
| *Day of incident* | Weekday | 42 (21%) | 125 (63%) | 167 (84%) | | Comparison group | | |
| | Weekend | 13 (7%) | 20 (10%) | 33 (17%) | 2,8 | 0,52 | 0,22–1,23 | 0,09 |
| *Time of incident* | Day shift (07h00-17h59) | 38 (19%) | 96 (48%) | 134 (67%) | | Comparison group | | |
| | Night shift (18h00-06h59) | 17 (9%) | 48 (24%) | 65 (33%) | 0,49 | 1,14 | 0,56–2,38 | 0,7 |

**Table 3. Statistical analysis of the five main incident types as categorised by patient safety type reported at KEH from April 2018 to March 2021.**

| Incident Type | Type of PSI | | Bivariate Analysis | | | Multivariate Analysis | |
|---|---|---|---|---|---|---|---|
| | No harm & near miss (n; %) | Harm (n; %) | Chi-square | P-value | Odds ratio (95% CI) | Odds ratio (95%CI) | P-value |
| Absconding | 22 (68.75) | 10 (31.25) | 32.51 | 0.00** | 0.11 (0.04–0.23) | 0.01 (0.02–0.05) | 0.00** |
| Clinical process not performed | 19 (40.43) | 28 (59.57) | 6.98 | 0.00** | 0.40 (0.19–0.85) | 0.19 (0.03–0.27) | 0.00** |
| Complications of surgery | 0 | 19 (100) | 7.96 | - | | Removed from model | |
| Falls | 2 (7.14) | 26 (92.86) | 6.77 | 0.00** | 5.77 (1.36–51.80) | 1.67 (0.14–5.73) | 0.92 |
| Pressure sores | 1 (9.09) | 10 (90.91) | 1.98 | 0.16 | 4 (0.54–176.72) | Removed from model | |

*level of significance at p <0.1

**level of significance at p <0.05

- cell counts zero

showed a similar pattern to those in the latter: no-harm and near-miss incidents comprised 52.73% [29/55] in the first half and 47.27% [26/55] in the latter half of the year. Harm incidents were 51.72% [75/145] and 48.28% [70/145] in the first versus the latter half of the year.

There were 83.50% [167/200] of incidents that occurred during a weekday and 16.50% [33/200] occurred over a weekend. A further breakdown based on the number of incidents that were reported by day, showed that most PSIs occurred on a Monday and Tuesday 19% [38/200]; with the least on a Saturday 7.50% [15/200] (Table 2). The highest harm incidents occurred on a weekday at 86.21% [125/145] versus 13.79% [20/145] over a weekend. No-harm and near-miss incidents were reported at 76.36% [42/55] during weekdays versus 23.64% [13/55] over the weekend.

The highest occurrence of PSIs was during the day shift at 67.50% [135/200]; and this was between 07h00 and 17h59. The night shift contributed to 32.50% [65/200] of the incidents (Table 3). Near-miss and no-harm incidents occurred at 69.09% [38/55] during the day shift versus 30.91% [17/55] at night; and 66.21% [96/145] of harm-related incidents were reported during the day shift versus the night at 33.79% [49/145].

There were no significant associations between type of PSI and day, time or month of occurrence.

## Nature of the incidents

The five highest incident types were clinical process/procedure [83/200; 41.50%], behaviour [49/200; 24.50%], falls [28/200; 14%], complications [20/200; 10%], and pressure sores [11/200; 5.50%].

Amongst the sub-categories of the incident type, the most frequently occurring safety incidents from the 'behaviour' category was absconding [32/200; 16%]; only 10 (31.25%) resulted in harm. In the 'clinical process/procedure' category the 'not performed when indicated' component was the highest [46/200; 23%], of which 27 (58.70%) resulted in harm. This was followed by pressure sores [11/200; 5.50%], of which there were 10 (90.91%) that were harm related. The next highest category was falls [28/200; 14%]; 26 (92.86%) resulted in harm and complication from a surgical operation [19/200; 9.50%], of which all were reported as harm incidents.

After bivariate analysis absconding, clinical process not performed, and falls, were significantly less likely to result in patient harm incidents. In the multivariate logistic regression absconding [OR 0.01, CI 0.02–0.05], and clinical process not performed [OR 0.19, CI 0.03–0.27] remained statistically significant at p <0.05 (Table 3). This indicates that both absconding and clinical process not performed incident types were less likely to be harm-related than a non-harm or near-miss type of PSI versus the other incident types.

## Contributory factors

A total of 252 contributory factors were reported for the 200 PSIs. Contributory factors were not mutually exclusive; and 18.50% [37/200] of incidents had more than one contributory factor. Staff factors rated as the highest contributor [121/252; 48.01%], followed by patient [97/252; 38.49%], environment [12/252; 4.76%], and organisation [22/252; 8.73%] (Table 4).

Contributing factors to no-harm and near-miss incidents were at 0.79% (2/252) for environmental factors and 3.97% [10/252] for organisational factors. Staff factors contributed 13.49% [34/252] and patient factors 10.32% [26/252]. In harm-related incidents environmental factors contributed 3.97% [10/252], organisational factors 4.76% [12/252], patient factors 28.17% [71/252]; staff factors were the highest at 34.52% [87/252].

Amongst staff factors, poor performance [99/252; 39.29%] and communication [15/252; 5.95%] were the highest-rated impeding factors. Patient factors contributing to incidents were mainly their pathophysiological state [53/252; 21.03%], followed by behaviour [28/252; 11.11%]. Poor or damaged infrastructure was the environmental-related contributory factor [12/252; 4.76%]. A paucity of protocols, policies, and procedures [9/252; 3.56%], and organisation of teams [9/252; 3.56%] contributed to organisational factors (Table 4).

## Outcomes

**Patient outcomes.** Patient outcomes were reported based on a scale of harm from none to death. The highest reported occurrence of harm was assessed as moderate at 29.50% [59/200], followed by mild at 22.50% [45/200], and severe harm in 11.50% [23/200] of incidents. No patient-related harm was noted in 27.50% [55/200] of the reports. Death as an outcome occurred in 9% [18/200] of incidents (Table 5).

**Organisational outcomes.** No organisational outcome attributed to a PSI was reported in 60% [120/200] of the incidents. The highest recognized adverse outcome was an increase in resources for the facility at 30.50% [61/200]. Damaged reputation occurred in 7.50% [15/200]; and a small percentage were noted to have caused property damage 1.50% [3/200], and a formal complaint was laid [1/200] (Table 5).

**Table 4. The contributory factors attributed to patient safety incidents from 1st April 2018 to 31st March 2021.**

| Variable | Total reported N = 252 (%) | Category (Total number reported) |
|---|---|---|
| *1. Staff* | 121 (48.01%) | 1.1 Behaviour [2] |
| | | 1.2 Cognitive [5] |
| | | 1.3 Communication [15] |
| | | 1.4 Performance [99] |
| *2. Patient* | 97 (38.49%) | 2.1 Behaviour [28] |
| | | 2.2 Cognitive [15] |
| | | 2.3 Emotional [1] |
| | | 2.4 Pathophysiological [53] |
| *3. Environment* | 12 (4.76%) | 3.1 Physical infrastructure poor/damaged |
| *4.Organisation* | 22 (8.73%) | 4.1 Lack of resources [9] |
| | | 4.2 Organisation of teams [4] |
| | | 4.3 Protocols/policies/procedures poor [9] |

**Table 5. The outcome factors attributed to patient safety incidents from 1st April 2018 to 31st March 2021.**

| Variable | Category | Total reported PSIs (N = 200) | Percentage |
|---|---|---:|---|
| *Patient Outcome* | None | 55 | 27.50 |
| | Mild | 45 | 22.50 |
| | Moderate | 59 | 29.50 |
| | Severe | 23 | 11.50 |
| | Death | 18 | 9.00 |
| *Organisational outcome* | None | 120 | 60.00 |
| | Increased resources required | 61 | 30.50 |
| | Damaged reputation | 15 | 7.50 |
| | Property damage | 3 | 1.50 |
| | Formal complaint | 1 | 0.50 |

## Discussion

This study showed that a total of 4.40 PSIs reported per 10 000 patient-days, or the percentage of admissions with at least one incident reported, was 0.43% over the three years. The frequency of reporting is low in comparison with other international studies.

A retrospective review of hospital medical records in eight LMICs, in which South Africa participated (Egypt, Jordan, Kenya, Morocco, Tunisia, Sudan and Yemen), indicated a lower frequency of adverse events at 8.2% of admissions (range 2.5–18%; n = 15 548) [15]. Locally, a retrospective cohort study undertaken during 2011 and 2016 at a public rehabilitative hospital within the eThekwini district showed similar reporting frequencies as this study. The study also used the voluntary reporting methods for analysis, and noted 4.12 PSIs per 10 000 patient days [16]. However, a retrospective observational study conducted between 2011 and 2014 at the same institution as this study, also using voluntary reporting, showed a lower reporting rate of 3.8 incidents per 10 000 patient days, indicating an increasing trend in the reporting rate for this facility [17].

An Italian and Canadian study compared teaching hospitals to community hospitals; and noted that the frequency of PSIs was higher at teaching hospitals. Reasons for this occurrence were attributed to patient factors, such as more acutely ill patients and more complex pathologies. Staff factors were miscommunication and lack of coordination from working with multidisciplinary teams for patient care [18, 19]. However, despite our study site being an academic tertiary facility, reporting rates were still low.

Reasons for the low reporting could be due to the inception of a new reporting system by the National Department of Health [20] in 2018, requiring an adjustment period for implementation. While the rates were the lowest in the year of implementation, they have progressively increased. This trend has been seen nationally and provincially, where rates have increased over the last three years. In the 2018/19 financial year KwaZulu-Natal reported 2447 PSIs; this increased to 6224 in the 2021/22 year [21]. The UK's National Reporting and Learning System which was developed in 2003 also witnessed an increase in reporting rates as healthcare workers became more familiar with the system and it was entrenched in their practice. The UK saw an exponential increase of 66 931 incidents reported for acute hospitals between 2003 and 2005 versus over 1 million in 2013 alone [22].

The improved reporting at the hospital is a combination of training as well as more focussed attention on reportion and management of adverse events in response to the increased medical litigation.

Additionally, the PSIs incident rates were based on voluntary submission of forms, and not based on medical chart audit reviews as noted by the European studies. The WHO PSI Reporting and Learning technical guidance report indicates that underreporting is common due in part to fear of retribution versus an opportunity for learning and growth [6].

## Patient safety incident type

The majority of PSIs were harmful incidents (72.50%), of which severe harm was noted in 11.50%.

The current study results in which the majority of PSIs were harm related, are more aligned with outcomes within LMICs in which the public healthcare sector suffers systemic constraints. A study conducted in Coronary Care Units in the eThekwini district using a self-administered questionnaire to collect data on types, frequency and patient outcomes of reported PSIs in CCUs and how they are managed showed that 47% of the adverse events reported in intensive care units were moderate, severe or catastrophic. The findings of our study show similarity in that 10% of the reported incidents were catastrophic. Compared to the 11,5% in the current study. In 2011 the WHO published a study in which South Africa participated with six Middle Eastern and another African country. This retrospective medical record review of over 18 000 patients from 26 hospitals showed that almost a third of patients in which harm was identified died, while 14% had a permanent disability, 16% moderate disability; and the remainder were classified as minimal (30%) or non-specified (8%). Most PSIs (80%) were noted to be preventable [15]. A study conducted in 2003 amongst gynaecology patients at the same site as this study, showed that 52% (n—220) of PSIs were deemed avoidable. Contrary to our findings, the majority were minor harm (disability lasting less than 6 months); however the mortality rate was higher, and occurred in 17.7% of incidents as an outcome [23]. This was a medical record review and was not based on voluntary reporting – it may have identified a greater number of minor-harm related incidents. The results in our study may be biased towards harm events; healthcare workers are more likely to report incidents with higher severity assessment codes.

Although, Absconding and Clinical process not performed were the most frequently reported PSI at the hospital, these two adverse events were less likely to be associated with severe harm. comprised acts of omission such as failure to examine a patient, to order a procedure, to perform preoperative preparation, or to follow protocols. These included missed injuries, delay in patients sent to theatre; or cancellation of theatre, and a failure to recognise difficult anaesthetic procedures.

patient factors 28.17% [71/252]; staff factors (34.52% [87/252]).were the highest contributory factors to harm related patient safety incidents. A large-scale medical-record review of patient safety incidents in the Middle East and Africa, which identified therapeutics errors as the most common (34.2%). These were acts of omission, failure to make a correct diagnosis, and a delay in diagnosis and treatment. This large-scale study highlighted inadequate training and supervision of staff as main underlying factors [15]. A significant lack of skilled doctors and nurses, delays in initiating clinical care due to an overburdened service, and a delay in inter-institutional transport [24] were reported as the root causes of obstetric complications.

## Study limitations

**Underreporting.** The proportion of reported PSIs is low compared with other studies. This may be due to the voluntary nature of the reporting system. The aim is to allow healthcare workers to identify and report incidents as part of a safety culture, without fear of retribution. Voluntary reporting is the method most often used to detect medical errors and patient safety

incidents. Limitations include underreporting, and are related to time constraints, lack of adequate reporting systems, fear of litigation, reluctance to report one's own errors, uncertainty of the clinical importance of the events, and the lack of changes after reporting.

**Selection bias.** This study relied on voluntary reporting, which may not reflect the true proportion of the incidence. The study included all inpatient reported PSIs during the time period. Data was triangulated with electronic records and PSI committee minutes to ensure that all known reported PSIs were included. Therefore, the study is not intended to be a true measure of safety, but rather to identify risk, and to highlight incidents and their contributing factors for intervention and mitigation.

**Information bias and data quality.** This study used secondary data that was collected as part of clinical care for the research. A national standardised PSI reporting tool was used as part of the routine reporting. The tool is mainly a tick chart of patient and incident characteristics, for ease of completion, with specific spaces left for further description of the incident. The PSI forms were completed as part of clinical processes by healthcare workers, which is a component of the clinical governance framework for the hospital. Inherent in the use of secondary data is the lack of accuracy.

Orientation is conducted for new staff at the beginning of the year. Such provides an overview of PSI reporting; however, no specific training is provided on use of the guidelines to employees. This may result in inaccurate completion of PSIs or inappropriate classification; and may also be attributed to the low reporting, staff being unaware of which incidents ought to be reported. This may also provide a reason for most incidents being reported on weekdays and during the day shift, when there is greatest supervision of senior staff.

Inter-observer variability can occur, as there is no formal training on completion of forms for all cadres of staff. However, this may have been reduced as the forms were reviewed by the operational manager of the unit, the head of department, the medical manager, and the quality assurance manager during the patient safety committee meetings. The forms are signed by senior staff once an incident is closed. In addition, retrospective record reviews have shown that inter-observer variability may be high, especially with regard to the judgements of root cause and preventability [25].

**Missing data.** The PSI form provided a user-friendly format and enabled most of the data to be completed, with minimal missing data noted. The areas identified that were poorly completed and had missing data were the patient pathology/diagnosis. This area was not included as part of the analysis. The form did not include the patient's length of stay, and first or recurrent admission; and this was also excluded from the analysis.

PSIs forms are paper based. However, since 2018, they have been electronically uploaded to the Department of Health website as part of the Ideal Hospital requirements [26]. This was useful, paper-based records being easily misplaced. The electronic PSI forms were a further source of triangulation of data. A previous study conducted during 2011–2014 at the same institution on adverse health events noted that 130 records were missing [17]. Since 2018, there may have been an improvement in data management.

**Generalizability.** This was a single-centre study of a tertiary-level hospital. The factors chosen for the study were routinely collected data that could be measured, and that may be reproducible in other facilities. The study may be generalised to other tertiary hospitals. However, as noted and described earlier, the referral pathway for this hospital is unique and must be considered. The main incident types, contributing factors and their sub-categories, were extracted and interpreted as per the standardised national guidelines for patient safety incident reporting in South Africa, aligned with the WHO guidelines. It is acknowledged that, while this is relevant to South Africa, it may differ in non-local settings. The interpretation may differ

across other studies due to the differences in definitions, inclusion and exclusion criteria, and patient populations.

## Conclusion

Although this study identified low PSI reporting rates over the three years, there has been an increasing trend in reporting. Harm incidents were the highest reported, which may reflect a reporting culture more inclined to bring awareness to incidents that have negative outcomes. To improve the reporting of patient safety incidents there is a need to create a safe reporting culture where healthcare workers or patients who report must not be victimized or punished. In addition there should be meaningful analysis of the data and lessons learned should be cascaded across the hospital. Use of electronic patient records and online reporting of PSIs in an integrated online digital platform will improve record-keeping, prevent redundancy, and will better facilitate data analysis. Furthermore, a systematic approach is required to address the health service and clinical risk factors.

## Supporting information

**S1 Data.**
(XLSX)

## Acknowledgments

The authors wish to thank the Quality Assurance Manager and staff from the Health Information Department at the hospital for assisting with the provision of data.

## Author Contributions

**Conceptualization:** Swabhavika Singh.

**Data curation:** Swabhavika Singh.

**Formal analysis:** Ozayr Mahomed.

**Investigation:** Swabhavika Singh.

**Methodology:** Swabhavika Singh, Ozayr Mahomed.

**Supervision:** Ozayr Mahomed.

**Writing – original draft:** Swabhavika Singh.

**Writing – review & editing:** Ozayr Mahomed.

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
