## [Decision Letter · Decision Letter 0]

11 Jul 2023

PONE-D-23-16739Nature and type of patient-reported safety incidents at a tertiary hospital in South Africa during the COVID-19 period (2018-2021)- A retrospective reviewPLOS ONE

Dear Dr. Mahomed,

Thank you for submitting your manuscript to PLOS ONE. After careful consideration, we feel that it has merit but does not fully meet PLOS ONE’s publication criteria as it currently stands. Therefore, we invite you to submit a revised version of the manuscript that addresses the points raised during the review process.

We look forward to receiving your revised manuscript.

Kind regards,

Sanjoy Kumer Dey, M.D

Academic Editor

PLOS ONE

Journal Requirements:

Reviewers' comments:

Reviewer's Responses to Questions

**Comments to the Author**

1. Is the manuscript technically sound, and do the data support the conclusions?

Reviewer #1: Yes

Reviewer #2: Yes

2. Has the statistical analysis been performed appropriately and rigorously? 

Reviewer #1: Yes

Reviewer #2: Yes

3. Have the authors made all data underlying the findings in their manuscript fully available?

Reviewer #1: Yes

Reviewer #2: No

4. Is the manuscript presented in an intelligible fashion and written in standard English?

Reviewer #1: Yes

Reviewer #2: Yes

5. Review Comments to the Author

Reviewer #1: According to WHO the burden of patient safety incidents (PSI) is disproportionately greater in low and middle income countries. Therefore, it is important for health system research to carry out studies on patient safety in those countries. The authors of the paper under review have made an analysis of totally 200 PSIs out of 46 707 adult hospital admissions from a tertiary-level academic hospital in the eThekwini district of KwaZulu-Natal in the Republic of South Africa. The article is well-written and deserves to be published in PLOS. There are, however, a few methodological issues that the authors should clarify:

- the PSIs were reported using the standardized forms of the South African National Guidelines for Patient Safety Reporting. These forms should be described, as well as the way of reporting (paper forms or electronic system)

- the distribution of reporting by medical doctors and nursing staff should be explained

- the education of the hospital staff for the PSI reporting should be opened. Is the improved reporting from 2018 to 2021 explained e.g. by improved training of the staff?

- were the patients of the hospital also able to file PSIs?

Reviewer #2: The authors retrospectively reported on patient-reported safety incidents at a tertiary South African hospital during COVID-19 period (2018 – 2021).

A significant limitation of this paper is reporting bias with very low reported incidents. The numbers are extremely low for a 454 684 inpatient facility in a Province in South Africa that struggles to deal with billions of rands of medical legal claims. This is highlighted by the low reporting rate of hospital acquired infections and medication adverse drug reactions are significantly underreported with only 5 and 3 reported in a 3-year period. This limits the conclusions that can be drawn from the data as including robust comparisons with other data.

Second limitation is that the data is retrospective and descriptive in its analysis. Multivariate analysis exploring predictors of patients outcomes would make the read more informative.

Suggestions:

The investigators should refocus the manuscript on the incidents that are less prone to reporting bias.

The demographics (Table 2) are of interest and correlations with severity of incidents will be informative.

Methods:

Significantly more information are required about the data source: what requires reporting, how is reporting done, analysis and verifying data when reported, root cause analysis, strategies to minimize reporting bias and encourage reporting etc. etc.

Data analysis:

Past tense & future tense used in the same paragraph.

Discussion:

Shorten the discussion.

There is no reflection on the South African context.

6. PLOS authors have the option to publish the peer review history of their article (what does this mean?). If published, this will include your full peer review and any attached files.

Reviewer #1: No

Reviewer #2: No

---

## [Author Response · Author response to Decision Letter 0]

2 Aug 2023

Reviewer ⌗ 1: 

1. The PSIs were reported using the standardised forms of the South African National Guidelines for Patient Safety Reporting. These forms should be described, as well as the way of reporting (paper forms or electronic system)

 We have added a description under the section Data Collection Tool and Procedures Line 83-103

2. the distribution of reporting by medical doctors and nursing staff should be explained The majority of the patient safety events were reported by nursing staff. 

3. the education of the hospital staff for the PSI reporting should be opened. Is the improved reporting from 2018 to 2021 explained e.g. by improved training of the staff? The improved reporting is a combination of training as well as more focussed attention by the National and Provincial Health Departments on the completion and reportion of adverse events in response to the increased medical litigation. (Line 315-317)

4. 

- were the patients of the hospital also able to file PSIs?

 Patients usually report any complaints or adverse events through the complaints mechanism at the hospital. 

Reviewer ⌗ 2: 

1. The investigators should refocus the manuscript on the incidents that are less prone to reporting bias Reviewers point acknowledged. It was our intention to provide the prevalence as reported, with the limitations. We will write a separate manuscript with focussed patient safety incidents. 

2. The demographics (Table 2) are of interest and correlations with severity of incidents will be informative We thank reviewer for the comments. It is our intention to submit. Second manuscript looking at bivariate and multivariate analysis using demographic variables

3. more information are required about the data source: what requires reporting, how is reporting done, analysis and verifying data when reported, root cause analysis, strategies to minimize reporting bias and encourage reporting Modifications made between line 87 and 109 

 Past tense & future tense used in the same paragraph Corrected

 There is no reflection on the South African context. Very few previous studies conducted in South Africa. Line 301-304 quote a previous study in the same institution.

---

## [Decision Letter · Decision Letter 1]

31 Aug 2023

PONE-D-23-16739R1Nature and type of patient-reported safety incidents at a tertiary hospital in South Africa during the COVID-19 period (2018-2021)- A retrospective reviewPLOS ONE

Dear Dr. Mahomed,

Thank you for submitting your revised manuscript. It has been observed that review comments not appropriately addressed. we feel that it has merit but does not fully meet PLOS ONE’s publication criteria as it currently stands. Therefore, we invite you to submit a revised version of the manuscript that addresses the points raised during the review process.

We look forward to receiving your revised manuscript.

Kind regards,

Sanjoy Kumer Dey, M.D

Academic Editor

PLOS ONE

Reviewers' comments:

Reviewer's Responses to Questions

**Comments to the Author**

1. If the authors have adequately addressed your comments raised in a previous round of review and you feel that this manuscript is now acceptable for publication, you may indicate that here to bypass the “Comments to the Author” section, enter your conflict of interest statement in the “Confidential to Editor” section, and submit your "Accept" recommendation.

Reviewer #1: All comments have been addressed

Reviewer #2: (No Response)

2. Is the manuscript technically sound, and do the data support the conclusions?

Reviewer #1: Yes

Reviewer #2: Yes

3. Has the statistical analysis been performed appropriately and rigorously? 

Reviewer #1: Yes

Reviewer #2: Yes

4. Have the authors made all data underlying the findings in their manuscript fully available?

Reviewer #1: Yes

Reviewer #2: No

5. Is the manuscript presented in an intelligible fashion and written in standard English?

Reviewer #1: Yes

Reviewer #2: Yes

6. Review Comments to the Author

Reviewer #1: The authors have responded to the comments of the reviewers and revised their manuscript appropriately. The manuscript is acceptable for publication.

Reviewer #2: The authors opted to dismiss the comment section below stating that a second manuscript will be submitted addressing the comments. Why not address it in this manuscript?

My original comments were:

A significant limitation of this paper is reporting bias with very low reported incidents. The numbers are extremely low for a 454 684 inpatient facility in a Province in South Africa that struggles to deal with billions of rands of medical legal claims. This is highlighted by the low reporting rate of hospital acquired infections and medication adverse drug reactions are significantly underreported with only 5 and 3 reported in a 3-year period. This limits the conclusions that can be drawn from the data as including robust comparisons with other data.

Second limitation is that the data is retrospective and descriptive in its analysis. Multivariate analysis exploring predictors of patients outcomes would make the read more informative.

Suggestions:

The investigators should refocus the manuscript on the incidents that are less prone to reporting bias.

The demographics (Table 2) are of interest and correlations with severity of incidents will be informative.

My second set of comments were:

Discussion:

Shorten the discussion.

There is no reflection on the South African context.

The shortening of the discussion was ignored.

The second comment had a dismissive response:

Very few previous studies conducted in South Africa. Line 301-304 quote a previous study in the same institution.

A 2 second Google review gave the following citation - and there will be more especially if adverse events are included in the search.

Gqaleni, T.M. & Bhengu, B.R., 2020, ‘Analysis of Patient Safety Incident reporting system as an indicator of quality nursing in critical care units in KwaZulu-Natal, South Africa’, Health SA Gesondheid 25(0), a1263. https://doi.org/10.4102/hsag.v25i0.1263

7. PLOS authors have the option to publish the peer review history of their article (what does this mean?). If published, this will include your full peer review and any attached files.

Reviewer #1: No

Reviewer #2: No

---

## [Author Response · Author response to Decision Letter 1]

17 Sep 2023

Reviewer ⌗ 2: 

1. The investigators should refocus the manuscript on the incidents that are less prone to reporting bias We have focussed on five main adverse events 

2. The demographics (Table 2) are of interest and correlations with severity of incidents will be informative Table 2 revised

 There is no reflection on the South African context. We have shortened the discussion.We have included the study suggested by the reviewer.

---

## [Editor Report · Decision Letter 2]

23 Oct 2023

Nature and type of patient-reported safety incidents at a tertiary hospital in South Africa during the COVID-19 period (2018-2021)- A retrospective review

PONE-D-23-16739R2

Dear Dr. Ozayr Mahomed,

We’re pleased to inform you that your manuscript has been judged scientifically suitable for publication and will be formally accepted for publication once it meets all outstanding technical requirements.

Kind regards,

Sanjoy Kumer Dey, M.D

Academic Editor

PLOS ONE

---

## [Editor Report · Acceptance letter]

26 Oct 2023

PONE-D-23-16739R2 

Nature and type of patient-reported safety incidents at a tertiary hospital in South Africa during the COVID-19 period (2018-2021)- A retrospective review 

Dear Dr. Mahomed:

I'm pleased to inform you that your manuscript has been deemed suitable for publication in PLOS ONE. Congratulations! Your manuscript is now with our production department. 

Kind regards, 

on behalf of

Dr. Sanjoy Kumer Dey 

Academic Editor

PLOS ONE